# Reinforcement Learning from Wild Animal Videos

**Wild Animal Videos**

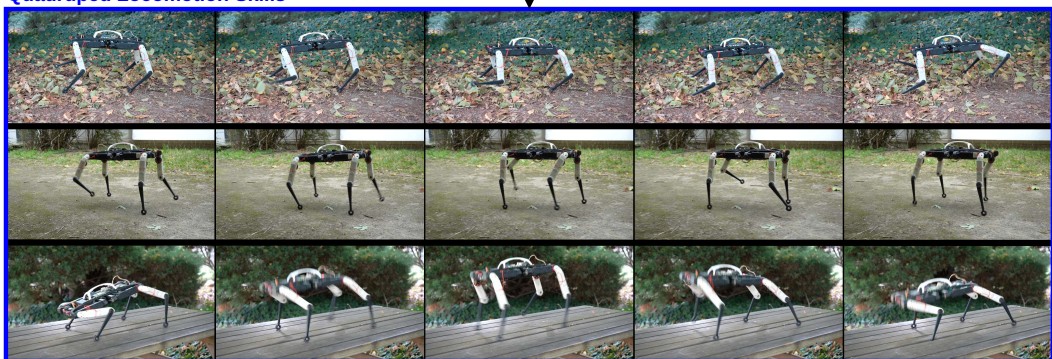

**Quadruped Locomotion Skills**

Figure 1: RLWAV trains a video classifier on 8,791 wild animal videos in natural environments to transfer the learned skills to a real quadruped robot with reinforcement learning, without relying on reference trajectories nor hand-designed rewards for each skill. Here, we present the motions resulting from the transfer of the keeping still, walking, and jumping skills to the Solo-12 robot.

## ABSTRACT

We propose to learn legged robot locomotion skills by watching thousands of wild animal videos from the internet, such as those featured in nature documentaries. Indeed, such videos offer a rich and diverse collection of plausible motion examples, which could inform how robots should move. To achieve this, we introduce Reinforcement Learning from Wild Animal Videos (RLWAV), a method to ground these motions into physical robots. We first train a video classifier on a large-scale animal video dataset to recognize actions from RGB clips of animals in their natural habitats. We then train a multi-skill policy to control a robot in a physics simulator, using the classification score of a third-person camera capturing videos of the robot's movements as a reward for reinforcement learning. Finally, we directly transfer the learned policy to a real quadruped Solo. Remarkably, despite the extreme gap in both domain and embodiment between animals in the wild and robots, our approach enables the policy to learn diverse skills such as walking, jumping, and keeping still, without relying on reference trajectories nor hand-designed rewards.

## 1 INTRODUCTION

Robot learning for control often involves a repetitive cycle: designing algorithms - such as refining a reward function in reinforcement learning (RL) -, analyzing the resulting behavior through video, and iterating until a satisfactory policy is achieved. Learning directly from videos could streamline this process by directly optimizing for visually successful rollouts, thus aligning the training process with the desired outcome. Internet videos offer a vast and diverse array of motion examples across a wide range of scenarios. Similar to how foundation models in computer vision and natural language processing have achieved impressive generalization by leveraging internet-scale data (Radford et al., 2021; Rombach et al., 2022; Wang et al., 2022; Achiam et al., 2023; Touvron et al., 2023), allowing robots to imitate these motions at scale could lead to new advancements in generalists robots. Just as humans and animals often learn by observing others (Molenberghs et al., 2009), even those with different body morphologies, robots could benefit from similar observational learning strategies. Recent research has demonstrated the potential of leveraging large datasets of videos of humans interacting with objects in everyday activites for robotic manipulation (Shao et al., 2021; Bahl et al., 2023; Bharadhwaj et al., 2024). In this work, we instead ask the following question: *Can robots learn locomotion skills by watching thousands of wild animal videos ?*

There exists a wealth of footage showcasing a wide range of animal species, from mammals and reptiles to amphibians and birds Feng et al. (2021); Ng et al. (2022); Chen et al. (2023). However, compared to learning manipulation tasks from human demonstrations, learning locomotion from these videos presents additional challenges. First, the embodiment gap between animals and robots is wider, as body dynamics play a lead role in defining locomotion and balance behavior. Prior works on behavior transfer from animals to robots has aimed to reduce this gap by tracking keypoints or estimating poses from videos (Peng et al., 2020; Bohez et al., 2022; Han et al., 2024), typically using recordings of animals with similar morphology to the target robot (e.g. transferring motions from dogs to quadruped robots) in controlled lab environments. Yet, this approach limits the range of available video resources and restricts robot designs to specific bio-inspired shapes for which data acquisition is practically feasible. On the other hand, many natural skills, like walking or jumping, emerge across species and can be easily identified by human observers despite these morphological differences. We argue that the embodiment gap between robots and animal species with similar forms is no greater than the gap between different classes in the animal kingdom, for instance between birds and mammals. Cross-embodiment relationships can be learned, but doing so would require large video datasets, encompassing diverse species in their natural habitats, where they freely exhibit their behaviors, yet resulting in poor camera angles, occlusions, or multiple animals in the same frame. Two main limitations prevent the direct application of this idea. First, there is no obvious correspondence between the body mechanics of animals and robots. Second, cross-embodiment visual imitation requires physical grounding - replicating locomotion skills captured from animal videos as faithfully as possible while adhering to the capabilities and limits of the physical morphology of the robot.

To overcome these challenges, we introduce *Reinforcement Learning from Wild Animal Videos* (*RL-WAV*), a method for grounding skill concepts from videos of animals in their natural environments into the behaviors of legged robots. First, we train a video encoder network using the Animal Kingdom dataset (Ng et al., 2022), a large and diverse collection of labeled animal videos spanning various species, sourced from the internet such as wildlife documentaries. This network learns to recognize actions directly from video pixels, namely keeping still, walking, running and jumping. By training on a wide range of scenarios and embodiments, we intend for the network to generalize to robot behaviors in a zero-shot manner, i.e without having been trained on robot videos. Then, we train a multi-skill policy in a physics simulator (Makoviychuk et al., 2021; Rudin et al., 2021) to ground these learned action concepts in realistic robot behaviors. Using constrained RL (Schulman et al., 2017; Kim et al., 2024; Chane-Sane et al., 2024b), we optimize the robot policy to maximize, as a reward, the classification score of the corresponding skill label obtained on videos of the robot movements captured from a third person view in simulation. We impose task-agnostic constraints related to the robot embodiments, similarly for each transferred skill. The physics simulator, along with the constraints, ensures that the robot behaviors remain physically plausible and transferable to real-world scenarios. Despite the significant gap in domain and embodiment between animals and robots, our approach successfully enables robots to acquire distinct skills corresponding to the considered action classes in the animal video dataset, without the need for reference trajectories or skill-specific reward functions.

We validate our approach in simulation as well as on a real Solo-12 quadruped robot , demonstrating the emergence and successful transfer of multiple skills which include keeping still, walking with two different styles and jumping on the spot (see Figure 1). To the best of our knowledge, this is the first demonstration of successful transfer from a large, diverse dataset of wild animal videos across various species to physical robot locomotion.

## 2 RELATED WORK

Over the past few years, training locomotion policies in physics simulators Todorov et al. (2012); Freeman et al. (2021); Makoviychuk et al. (2021) using reinforcement learning, then transferring them to real robots (Peng et al., 2018b; Lee et al., 2020; Fu et al., 2021; Rudin et al., 2021; Li et al., 2023d; Aractingi et al., 2023) has proven remarkably effective for acquiring diverse skills for legged robots. These include high-speed running (Bellegarda et al., 2022; Margolis et al., 2024; He et al., 2024; Yang & Hwangbo, 2024), jumping (Bellegarda et al., 2020; Margolis et al., 2021; Li et al., 2023d; Smith et al., 2023; Zhang et al., 2024), and traversing challenging terrains Miki et al. (2022); Hoeller et al. (2022); Agarwal et al. (2023); Yang et al. (2023c); Zhuang et al. (2023); Hoeller et al. (2024); Caluwaerts et al. (2023); Zhuang et al. (2023); Hoeller et al. (2024); Chane-Sane et al. (2024a); Luo et al. (2024). Recently, constrained reinforcement learning has further simplified this process and enhanced its effectiveness (Kim et al., 2024; Lee et al., 2023; Chane-Sane et al., 2024b;a). In this work, we follow this sim-to-real approach with constrained RL.

To obtain more natural movements, some works propose to imitate reference trajectories (Peng et al., 2018a; Li et al., 2023a;b;c) or use them as style priors for downstream locomotion tasks (Peng et al., 2021; Escontrela et al., 2022; Yang et al., 2023b). In the context of animal imitation, these reference trajectories can be obtained from animal with close resemblance to the target robot, where typically dogs are employed to generate the reference motion via motion capture or by pose estimation from videos for quadruped robots (Peng et al., 2020; Bohez et al., 2022; Yao et al., 2022; Zhang et al., 2023; Li et al., 2023a; Han et al., 2024). This approach requires careful alignment between animals and target robots. We instead seek to scale the transfer across hundreds of species in the wild.

Learning cross-embodiment policies has been explored previously (Huang et al., 2020; Salhotra et al., 2023; Feng et al., 2023; Devin et al., 2017; Padalkar et al., 2023; Yang et al., 2023a; Shah et al., 2023; Shafiee et al., 2024; Yang et al., 2024) to learn policies across robot morphologies. Another line of work seeks to imitate human videos, typcally by extracting poses Qin et al. (2022); Mandikal & Grauman (2022); Shaw et al. (2023); Chen et al. (2024); Shaw et al. (2024) or keypoints(Peng et al., 2018c; Xiong et al., 2021; Bahl et al., 2022; Heppert et al., 2024) from videos and tracking the resulting reference motions. Explicitly extracting this intermediate representation often require careful alignment between videos and robots(Smith et al., 2019; Xiong et al., 2021; Zakka et al., 2021; Xu et al., 2023; Wang et al., 2023), which limits the use of a large portion of available video resources, including footages of animals in the wild. Instead, we show that it is possible to avoid this careful alignment and make use of videos captured in uncontrolled environments.

Leveraging large-scale human video datasets has been explored in robot learning to pretrain policies (Nair et al., 2022; Xiao et al., 2022; Ma et al., 2022; Majumdar et al., 2023; Bahl et al., 2023; Seo et al., 2022; Radosavovic et al., 2023; Ma et al., 2023; Mendonca et al., 2023; Ze et al., 2024) and augment RL (Schmeckpeper et al., 2020; Fan et al., 2022; Alakuijala et al., 2023). More closely related to our work, Shao et al. (2021); Chen et al. (2021); Chane-Sane et al. (2023) propose to employ video classifiers from human manipulation videos as the sole task reward function for training manipulation tasks. In this work, we adopt a similar approach based on learned video classifiers, but extend it to demonstrate that cross-embodiment transfer can scale to learning locomotion skills from wild animal videos across diverse species — a challenge that requires more visual generalization and extensive physics grounding.

## 3 METHOD

### 3.1 PROBLEM FORMULATION

Our goal is to learn a multi-skill locomotion policy for controlling a quadruped robot in simulation with RL, then transfer it to a real robot. To this end, we consider an infinite discounted constrained

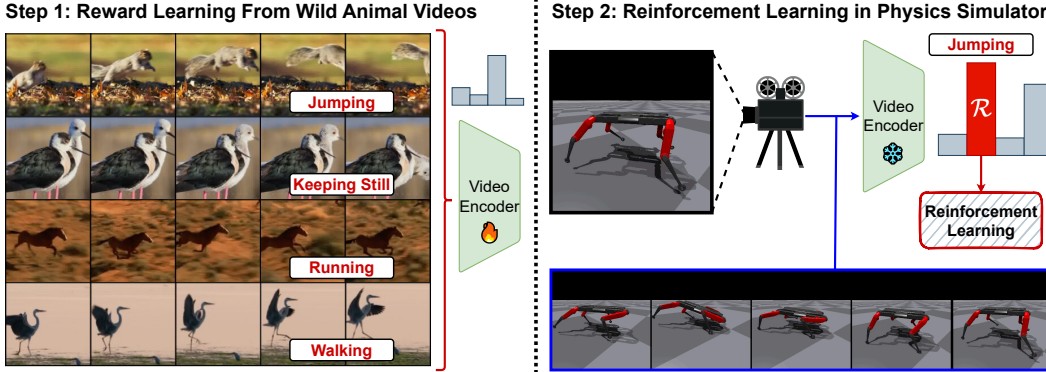

Figure 2: (Left) We train a video classifier to recognize actions from the Animal Kingdom dataset (Ng et al., 2022). (Right) With a third-person camera capturing videos of the robot's movement, we use the classification score for the desired skill as a reward to train the policy with RL.

Markov Decision Process $\left(\mathcal{S}, \mathcal{A}, \mathcal{R}, \gamma, \mathcal{T}, \{\mathcal{C}^i\}_{i \in I}\right)$ with state space $\mathcal{S}$, action space $\mathcal{A}$, discount factor $\gamma$, and stochastic reward $\mathcal{R} : \mathcal{S} \times \mathcal{A} \times \mathbb{R} \to \mathbb{R}^+$, dynamic $\mathcal{T} : \mathcal{S} \times \mathcal{A} \times \mathcal{S} \to \mathbb{R}^+$ and constraints $\{\mathcal{C}^i : \mathcal{S} \times \mathcal{A} \times \mathbb{R} \to \mathbb{R}^+\}_{i \in I}$. We defined $\mathcal{T}$, $\mathcal{R}$ and $\mathcal{C}^i$ in a stochastic manner to account for their partial observability. Constrained RL aims to find a policy $\pi : \mathcal{S} \to \mathcal{A}$ that maximizes the discounted sum of future rewards:

$$\max_{\pi} \mathbb{E}_{\substack{a_t=\pi(s_t), r_t \sim \mathcal{R}(.|s_t,a_t) \\ s_{t+1} \sim \mathcal{T}(.|s_t,a_t)}} \left[ \sum_{t=0}^{\infty} \gamma^t r_t \right], \tag{1}$$

while satisfying the constraints under the discounted state-action policy visitation:

$$\mathbb{P}_{(s,a) \sim \rho_\gamma^{\pi, \mathcal{T}}, c^i \sim \mathcal{C}^i(.|s,a)} \left[ c^i > 0 \right] \le \epsilon_i \; \forall i \in I. \tag{2}$$

Observations $s = (s^{\text{proprio}}, y) \in \mathcal{S}$ consist in proprioceptive measurements of the robot $s^{\text{proprio}}$ and a skill command $y$, such that by specifying different skill command to the policy, the user can control the robot to perform different skills. Actions $a \in \mathcal{A}$ produced by the policy are target joint angles for each joint of the robots that are converted to torques by a proportional-derivative controller operating at a higher frequency than the policy before being applied to the joints of the robot.

In learning-based locomotion, rewards and constraints are typically manually designed for each skill $y$. However, this process is often tedious and time-consuming, especially for legged locomotion, where we must balance task success with safe, physically plausible movements. Instead, we propose learning a multi-skill reward function $\mathcal{R}$ from animal videos to capture abstract locomotion concepts and employ constraints $\{\mathcal{C}^i\}_{i \in I}$ for physical grounding. These constraints are applied uniformly across skills to facilitate policy learning and enable effective and safe sim-to-real transfer.

### 3.2 LEARNING A MULTI-SKILL REWARD FUNCTION FROM WILD ANIMAL VIDEOS

**Action Recognition for RL** We train a video classifier to recognize actions from a large dataset of labeled videos of animals in the wild as our multi-skill reward function $\mathcal{R}$. This approach is grounded in two key observations: First, despite significant differences in morphology, humans can reliably recognize locomotion skills across vastly different species based solely on observation. For instance, even though spiders have more legs and are much smaller than humans, we can still identify when a spider is walking, jumping, or remaining idle. By leveraging recent advances in video action recognition, we anticipate that modern video classifiers can similarly develop an understanding of diverse animal behaviors, regardless of embodiment. Second, from a practical perspective, designing a control system for locomotion often involves iterative cycles: designing or modifying the algorithm, analyzing the resulting behaviors—typically through video generation in simulation or real robot observation—and refining the approach until a satisfactory policy is achieved. Integrating a neural network capable of interpreting robot behavior directly from video could streamline this process, aligning it more closely with the ultimate goal of achieving optimal locomotion.

**Animal Video Dataset**  We consider the Animal Kingdom dataset (Ng et al., 2022), a comprehensive collection of labeled videos capturing animals in their natural environments. The dataset consists of approximately 30,000 videos representing 850 species, including mammals, reptiles, birds, amphibians, fish, and insects (see Figure 1 and Appendix C). It features multi-label annotations of animal behaviors across 140 classes. While some of these classes, such as "Walking" and "Keeping Still," are directly relevant for transfer to our quadruped robot, others, such as "Flying," "Spitting Venom," or "Carrying In Mouth," are not applicable. In many cases, videos are annotated with multiple labels, reflecting complex behaviors—an animal might be walking while carrying a prey in its mouth, for example. We focus on four key classes of interest: "Keeping Still," "Walking," "Running," and "Jumping". We filter the dataset to include only videos containing one of these labels, discarding the other labels when a video contains multiple labels. Furthermore, we remove videos where more than one of the four selected behaviors occurred simultaneously. As a result, we curate a single-label dataset $\mathcal{D}^{\text{animal}} = \{(x^{\text{animal}}, y)_j\}_j$ comprising $8,791$ videos, where a video $x^{\text{animal}}$ is a sequence of $T$ images $(x_1^{\text{animal}}, x_2^{\text{animal}}, .., x_T^{\text{animal}})$ with label $y$ belonging to one of the four classes.

**Video Classifier**  We train a video classifier $f_\theta(x^{\text{animal}}, y)$ on the animal video dataset using the cross-entropy loss:

$$\mathcal{L}(\theta) = \mathbb{E}_{(x^{\text{animal}}, y) \sim D^{\text{animal}}} \left[ -\log f_\theta(x^{\text{animal}}, y) \right] \tag{3}$$

We parametrize $f_\theta$ with a Uniformer (Li et al., 2022), an efficient and light-weight architecture for video classification, to regress the probability distribution over the 4 classes from the animal video inputs. We chose the Uniformer over practical considerations for compute efficiency, although any architecture could work in principle. To improve out-of-distribution generalization, we also use random convolution augmentations (Lee et al., 2019) and model soups (Wortsman et al., 2022; Rame et al., 2022). Additional implementation details are given in Appendix B.1

## 3.3 Reinforcement Learning from Wild Animal Videos

**Reward**  Figure 2. illustrates our approach for training the policy using the learned video classifier as a reward function. In the simulator, we position a third-person camera to observe the robot. This camera tracks the robot's movement in 3D space and around the yaw axis. The camera captures $128 \times 128$ RGB images of the robot every 5 time steps, storing the previous frames in memory. These frames are then combined to form an 8-frame video sequence $x^{\text{robot}}$, which is fed into the video classifier. The classification score corresponding to the skill command input $y$ is used to construct our video-based reward function:

$$\mathcal{R}(s_t, a_t) = \begin{cases} f_\theta(x^{\text{robot}}, y) & \text{on image generation steps} \\ 0 & \text{otherwise} \end{cases}. \tag{4}$$

Rewards are assigned only at the time steps when the camera captures an image, while at all other time steps, a reward of zero is given. This approach avoids generating images at every time step for two reasons: (1) generating images is computationally expensive in the simulation, and (2) the video classifier was trained on animal videos at a lower frame rate than the simulation. Note that the robot does not observe the videos of itself that are used to compute the reward; these videos are only utilized during the training phase within the simulation. Additionally, the policy does not have access to a history of states, nor does it know the specific time steps when the camera captures images. Despite these partial observabilities, we found that the policy was still able to learn effectively.

**Constraints**  To ensure the physical grounding of animal locomotion skills, facilitate policy learning and enable effective sim-to-real transfer, we incorporate a set of constraints as $\mathcal{C}^i{}_{i \in I}$. These constraints include limiting joint angles, velocity, acceleration, and torque, while also imposing a minimum air time for the feet and restricting the roll orientation of the robot. Note that these constraints are independent of the skill command $y$.

**Policy Learning**  We use PPO (Schulman et al., 2017) as our RL optimizer. We use CaT (Chane-Sane et al., 2024b) to learn a policy that comply with the constraints. To facilitate policy learning, we use an additional symmetry loss (Yu et al., 2018; Abdolhosseini et al., 2019). The differences between the learned locomotion skills arise solely from the learned reward function. Additional implementation details are given in Appendix B.2.

Table 1: Influence of the learned video-based reward function on the policy

| Method | Keeping still | | Walking | | Running | | Jumping | |
|---|---|---|---|---|---|---|---|---|
| | $\|vel_{xy}\| \downarrow$ | Style | $vel_x \uparrow$ | Style | $vel_x \uparrow$ | Style | $\Delta z \uparrow$ | Style |
| RLWAV | 12.3 (1.1) | 1.0 | 23.0 (15.4) | 0.88 | 35.1 (12.9) | 1.0 | 18.5 (3.5) | 0.88 |
| no soup | 14.2 (1.2) | 1.0 | 21.8 (13.9) | 0.62 | 26.0 (17.9) | 0.75 | 10.6 (8.1) | 0.38 |
| no curating | 8.9 (2.3) | 1.0 | -0.1 (1.4) | 0.0 | -0.9 (5.7) | 0.12 | 19.1 (2.0) | 0.75 |

## 4 EXPERIMENTS

### 4.1 EXPERIMENTAL SETUP

The policies are trained in the IsaacGym simulator using massively parallel environments (Makoviy-chuk et al., 2021; Rudin et al., 2021). A policy can be trained with RL on a single NVIDIA RTX 4090 GPU in less than 4 hours. While this corresponds to a similar simulation time compared to (Rudin et al., 2021), it amounts to a much higher wall-clock time due to the frequent rendering of images. After training in simulation, we directly deploy the policy on a real Solo-12 quadruped robot. The policy runs at 50Hz on a Raspberry Pi 5. Target joint positions are sent to the onboard proportional-derivative controller running at 10kHz. The user commands the desired skill to the robot through a joystick.

**Evaluation protocol**   To quantitatively evaluate our approach and compare different policies, we propose separate evaluation metrics for each skill:

- Keeping still: we measure the average of the absolute velocity in any direction $|vel_{xy}|$ in cm/s (lower is better)
- Walking and Running: we measure the average velocity in the forward direction of the robot $vel_x$ in cm/s (higher is better)
- Jumping: we measure the average vertical displacement $\Delta z$ in cm (higher is better)

In addition, we qualitatively evaluate the style of each policy by rendering a video of a policy rollout in simulation and manually rating the quality of the policy. We grade the videos according to the following scale: we give 1.0 if the movement is perfectly identifiable as correct, 0.5 if the movement is close but not entirely accurate (for example, walking on the spot without moving forward), and 0.0 if the movement is unrecognizable or ambiguous with respect to another skill.

**Baselines and ablations**   To highlight the importance of our design choices, we compare our approach, *RLWAV*, to the following ablations:

- *no curating*: we train the video classifier on the full multi-label video dataset with binary cross-entropy loss instead of our chosen single-label subset.
- *no soup*: we don't use model soup to train the video classifier.
- *no sym. loss*: we remove the symmetry loss function during RL.
- *update 8*: we update the third-person video every 8 steps instead of 5 during RL.
- *low pose*: we set the base pose of the robot closer to the ground during RL.
- *Camera {1,2,3,4}*: we try four alternative positions for the third-person camera during RL.

For each experiment, we report the mean and standard deviation over 4 RL training seeds.

### 4.2 SIMULATION EXPERIMENTS

**Skill transfer from wild animal videos to robots**   Table 1 presents the performance results of our approach across four skills. Despite the absence of a skill-specific reward function or predefined reference trajectory, we observe the emergence of distinct, recognizable behaviors for each skill (see Figure 4). For the "keeping still" task, the robot successfully learns to remain stationary, although

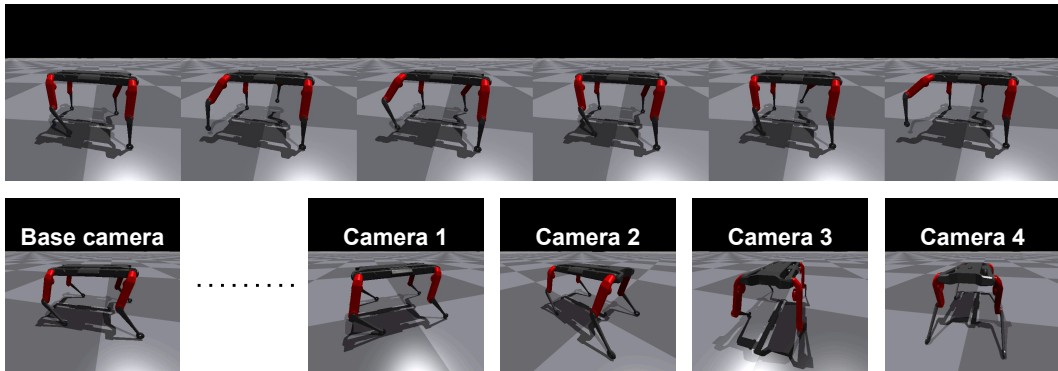

Figure 3: (Top) Example of failure case for the "walking" skill, where the robots performs walking motions for only two legs without moving forward, fooling the video reward function.
(Bottom) Base camera in nominal setting, camera 1,2,3,4 in the ablation study.

it consistently exhibits minor leg movements. In the walking and running tasks, the Solo12 robot adopts a trotting motion in the forward direction. Although the running task is slightly faster than the movements generated for "walking", the policy does not generate proper flying phases (i.e. with the robot having no ground contact during some movement phases) but mostly broader limb movements and slight yaw rotations of the base. When issued a jumping command, the robot performs broad, rhythmic pumping motions, either in place or with slight drift, often entering phases where none of its feet are in contact with the ground. These results validate our approach ability to effectively transfer motion skill concepts learned from wild animal videos to quadruped robot locomotion.

**Effect of video classifier training on skill acquisition** In Table 1, we also analyze how different training protocols for the video classifier impact the downstream policy learning by comparing our approach against *no soup* and *no curating*. Removing model soup (*no soup*) leads to a slight decrease in performance across all locomotion skills, with the most pronounced effect observed in jumping. We attribute this to the improvements in out-of-distribution generalization provided by model soup, which aids in transferring the video classifier to the robot domain and delivering more relevant reward signals for policy learning. Moreover, without our data curation strategy (*no curating*), the robot is unable to learn walking and running. We attribute this to the fact that our data curation process tailors the reward signals more specifically to the target skills.

**Policy learning ablations** In Table 2, we analyze the impact of various components in our policy learning method through ablation studies. Removing the symmetry loss function (*no sym.*) significantly reduces the robot's performance in the running skill. This is likely due to the single camera setup, which makes it difficult for the robot to learn the symmetric leg movements typically required for achieving forward motions. Increasing the number of steps between camera renderings(*update 8*), and consequently delaying reward updates, results in performance drops across walking, running, and jumping tasks. We attribute this decline to the sparser reward signals, which hinder effective learning. The impact is particularly pronounced for jumping, likely because the dynamic nature of the motion requires finer temporal resolution for better reward feedback. Additionally, altering the base pose of the robot to be lower to the ground with more flexed legs (*low pose*) prevents the effective learning of walking and running skills. In this setting, the robot performs jumping motions instead of running whereas the robot's score in the jumping task improves under this condition. This may be because the lower pose facilitates jumping, making it easier for the robot to discover and exploit jumping motions, which still yield positive rewards from the video classifier. This aligns with the observation that certain walking and running behaviors in animals, for example cheetahs, can resemble a forward-jumping motion. Moreover, effective walking movements are not learned; instead, the robot often appears to walk in place without moving forward. Figure 3 (top) illustrates one such failure case, where the robot fails to use all its legs properly.

**Effect of the camera position** In Table 2, we examine how the camera position used to capture videos of the robot in simulation affects policy learning by comparing policies learned from four different camera placements relative to the robot, as shown in Figure 3 (bottom). As the camera po-

Table 2: Ablation of the policy learning and camera position

| Method | Keeping still | | Walking | | Running | | Jumping | |
|---|---|---|---|---|---|---|---|---|
| | $|vel_{xy}| \downarrow$ | Style | $vel_x \uparrow$ | Style | $vel_x \uparrow$ | Style | $\Delta z \uparrow$ | Style |
| RLWAV | 12.3 (1.1) | 1.0 | 23.0 (15.4) | 0.88 | 35.1 (12.9) | 1.0 | 18.5 (3.5) | 0.88 |
| no sym. loss | 15.3 (1.1) | 1.0 | 22.9 (20.7) | 0.5 | 9.3 (25.7) | 0.5 | 17.6 (2.3) | 1.0 |
| update 8 | 13.2 (0.6) | 1.0 | 17.1 (10.1) | 0.75 | 19.4 (3.2) | 0.88 | 5.7 (2.8) | 0.0 |
| low pose | 12.4 (0.7) | 1.0 | -3.2 (7.0) | 0.38 | 2.3 (21.8) | 0.12 | 22.6 (1.5) | 1.0 |
| Camera 1 | 12.8 (1.4) | 1.0 | 46.4 (16.5) | 1.0 | 36.8 (40.7) | 0.75 | 19.6 (1.4) | 1.0 |
| Camera 2 | 15.1 (0.6) | 1.0 | 11.0 (18.9) | 0.62 | 40.1 (5.4) | 0.75 | 17.6 (5.4) | 1.0 |
| Camera 3 | 13.7 (1.4) | 1.0 | 42.7 (12.8) | 0.88 | 6.8 (3.6) | 0.5 | 7.0 (0.7) | 0.0 |
| Camera 4 | 11.9 (2.2) | 1.0 | -4.3 (3.7) | 0.0 | 37.9 (4.0) | 0.75 | 11.5 (3.4) | 0.62 |

sitions progress from camera 1 to camera 4, the views become more extreme, making it increasingly difficult for the video classifier to provide accurate reward feedback to the policy. Unsurprisingly, policies learned from more optimal camera positions, where the robot's full body is more visible (cameras 1 and 2), perform better across most skills compared to policies learned from more extreme camera angles (cameras 3 and 4). The exception is the "keeping still" skill, which is successfully learned from all camera positions. Interestingly, certain camera angles benefit specific skills while impairing others. For example, cameras positioned more towards the back of the robot (cameras 1 and 3) result in better performance for walking, whereas cameras positioned towards the front (cameras 3 and 4) lead to improved performance in running.

## 4.3 REAL-WORLD EXPERIMENTS

We deploy RLWAV directly onto our real Solo-12 platform (Grimminger et al., 2020). Figures 1 and 5 show policy rollouts for the keeping still, walking, running and jumping skills, where the policy was tested in outdoor environments. The results on the real robot are best demonstrated in the supplementary video accompanying this submission. The policy learned in simulation transfers successfully to the real robot. The "keeping still" skill functions as expected, and for the jumping task, the robot consistently jumps in place. Additionally, when commanded to perform walking or running, the robot trots forward. While the policy learned a walking gait with relatively straight legs and a high body posture—less stable than traditional walking poses for this kind of quadruped robots—the robot still manages to walk on uneven outdoor terrains. However, similar to the simulation results, the walking and running skills appear alike on the real robot. The running skill shows a slight increase in speed and slightly broader movements of the body with increased slippage in particular for the rear foot. The running speed is limited by the feet sliding on the ground, which is a sim-to-real artifact. Lastly, despite the policy not being explicitly trained for skill command switching in simulation, it smoothly transitions between skills when commanded by the user in the real-world deployment.

## 5 CONCLUSION

We introduced RLWAV, a method for grounding natural motions learned from thousands of wild animal videos into physical robots. Through extensive experiments in simulation and successful deployment on a real Solo-12 robot, we demonstrated the extreme cross-embodiment transfer of behaviors from animal videos to a physical quadruped robot, enabling it to perform various skills such as keeping still, jumping, walking, and, to a lesser extent, running. Our results showcase the potential of large-scale datasets of internet videos for legged locomotion.

While promising, the behaviors we achieved still lag behind the state-of-the-art in learning-based locomotion. In this work, we repurposed a video dataset originally designed for wildlife behavior understanding, extracted motions using conventional video classification techniques, and applied a standard on-policy RL algorithm. To scale to larger video datasets, broaden the range of locomotion skills, and achieve more agile and precise behaviors, future work could explore curating video datasets specifically aligned with control, using more advanced video understanding techniques, and designing policy learning algorithms that better leverage the structure of video-based robot learning.

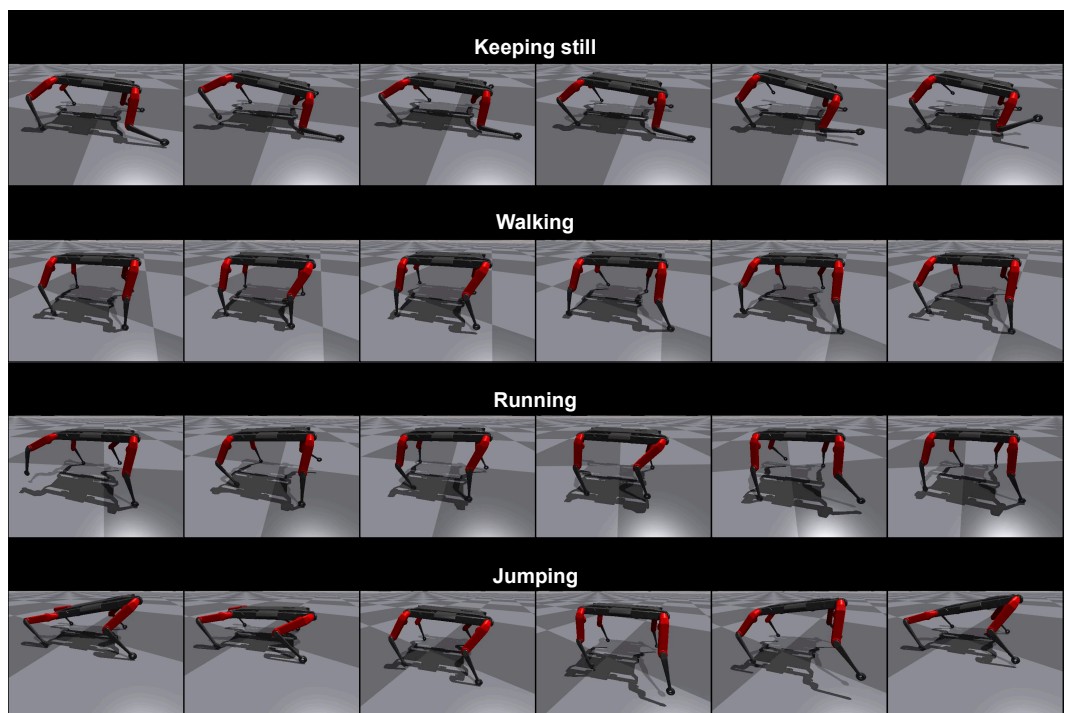

Figure 4: Rollout examples in simulation for the 4 skills considered.

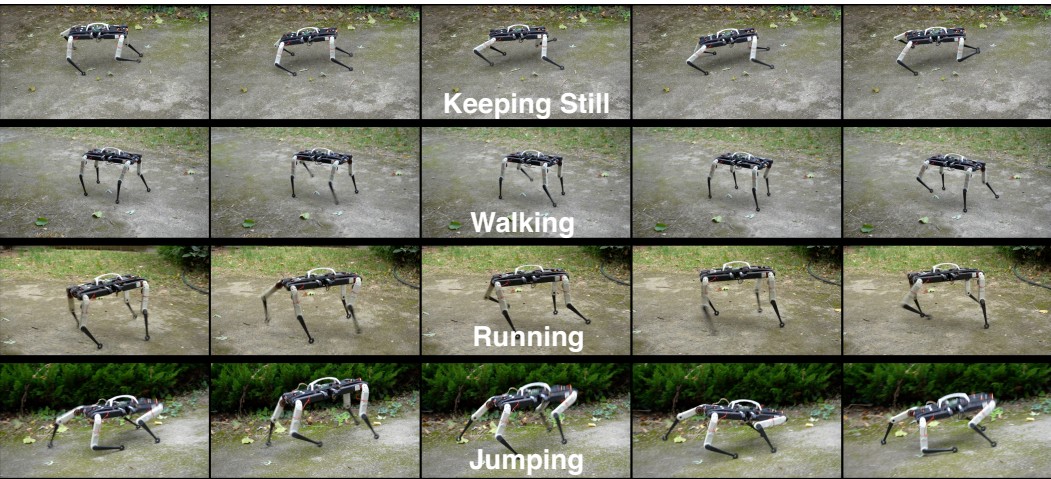

Figure 5: Rollout examples for the 4 skills considered on the real Solo-12. The results on the real robot are best demonstrated in the supplementary video accompanying this submission.

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

## A ADDITIONAL RESULTS

In Figure 6, we analyze the effect of varying the number of training videos on the emergence of downstream locomotion skills. The video classifier is trained on random subsets of the dataset described in Section 3.2. Keeping only 50% of the animal videos does not hinder the emergence of the desired locomotion skills. However, further reducing the dataset to 25% or 12.5% impairs the emergence of "Keeping Still" and "Running." These results highlight the importance of training the video-based reward on a sufficiently large and diverse dataset to enable effective generalization to robot locomotion.

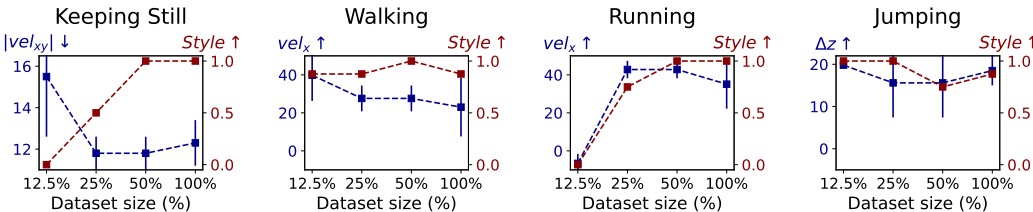

Figure 6: Impact of animal video dataset size (% of total videos) on skill emergence.

In Table 3, we evaluate the effect of removing specific constraints during policy learning on the development of locomotion skills. Eliminating the base orientation constraint around the roll axis has minimal impact, except in "Running," where the robot sometimes exhibits exaggerated movements that fail to effectively propel it forward. In contrast, removing the foot air-time constraint severely impairs "Walking" and "Running." The robot demonstrates leg motions, but these fail to translate into forward movement, instead resulting in slipping, high-frequency ground contacts, and insufficient foot lifting (see supplementary video). This observation aligns with prior research on learning-based locomotion which also employ foot air-time constraints to avoid exploiting the limitations of the physics simulator Rudin et al. (2021); Chane-Sane et al. (2024b).

Table 3: Ablation on policy learning without the base orientation constraint around the roll axis and without the foot air-time constraint.

| Method | Keeping still | | Walking | | Running | | Jumping | |
|---|---|---|---|---|---|---|---|---|
| | $|vel_{xy}|\downarrow$ | Style | $vel_x\uparrow$ | Style | $vel_x\uparrow$ | Style | $\Delta z\uparrow$ | Style |
| RLWAV | 12.3 (1.1) | 1.0 | 23.0 (15.4) | 0.88 | 35.1 (12.9) | 1.0 | 18.5 (3.5) | 0.88 |
| w/o orientation | 12.8 (1.2) | 1.0 | 42.8 (6.2) | 1.0 | 32.5 (19.0) | 0.75 | 18.2 (5.1) | 0.75 |
| w/o air time | 7.3 (0.3) | 1.0 | 4.9 (3.7) | 0.12 | -3.5 (9.0) | 0.25 | 19.0 (2.3) | 1.0 |

## B IMPLEMENTATION DETAILS

### B.1 ANIMAL VIDEO CLASSIFICATION

We finetune a Uniformer-S (Li et al., 2022) video encoder pretrained on Kinetics 400 (K400) (Kay et al., 2017). We adopt AdamW optimizer (Loshchilov et al., 2017), MixUp (Zhang, 2017) and a linear warmup then cosine annealing learning rate scheduler. We use random convolutions (Lee et al., 2019) 95% of the time, where the same random convolution kernel is applied to the whole video sequence. We train the video encoder with a batch size of $64$, a learning rate of $3e-5$, $8$ frames per video and sampling stride of $4$. We first train only a classifier head on top of the K400 pretrained Uniformer-S backbone. Starting from these weights, we then fully finetune 12 different models by varying the stochastic depth rate in $\{0.3, 0.4\}$, the weight decay in $\{0.05, 0.01, 0.1\}$ and total epochs in $\{30, 50\}$ between the runs. Finally we uniformly average the weights of these models (Wortsman et al., 2022; Rame et al., 2022).

## B.2 REINFORCEMENT LEANING

We train the policy in simulation using Proximal Policy Optimization (PPO) (Schulman et al., 2017). We use a custom PPO implementation based on CleanRL (Huang et al., 2022). The policy is parameterized by a multilayer perceptron with hidden dimensions $[512, 256, 128]$ and elu activations. We use an additional symmetry term to the policy loss during RL (Yu et al., 2018):

$$\mathcal{L}_{\text{symmetry}}(\phi) = \|\pi_\phi(s) - \text{sym}(\pi_\phi(\text{sym}(s)))\|_2^2, \tag{5}$$

where the sym operation applies a symmetry transformation to the state or action based on the left-right symmetry of the quadruped robot.

We use 2048 parallel environments in the Isaac Gym simulator (Makoviychuk et al., 2021). Skill commands are evenly sampled across the actors. During an episode, the skill command remain fixed as we didn't find any problem transitioning between skills during deployment, see Section 4.3. We generate RGB images in simulation at $128 \times 128$ resolution every 5 RL steps, storing previous frames. On rendering steps, we then compute the rewards from the 8-frames robot videos $x^{\text{robot}}$ using the learned video classifier. Because rewards for different skills may have different magnitudes, we use the following reward formulation:

$$\mathcal{R}(s_t, a_t) = \begin{cases} \alpha_y f_\theta(x^{\text{robot}}, y) + \beta_y & \text{on image generation steps} \\ 0 & \text{otherwise,} \end{cases} \tag{6}$$

where $\alpha_y$ and $\beta_y$ are adaptively optimized for each skill based on the reward statistics across the actors to normalize the rewards between $0$ and $1$ throughout training.

We use Constraints as Terminations (CaT) (Chane-Sane et al., 2024b) to enforce the constraints $\{\mathcal{C}^i\}_{i \in I}$ during RL. Table 4 lists all the constraints, where knee and base collision constraints and foot contact force constraints are applied as hard constraints whereas the other constraints are applied as soft constraints in the CaT framework. These constraints restrict the behavior search space of RL to facilitate policy learning and enable effective and safe sim-to-real transfer. Importantly, these constraints are applied independently regardless of the skill command $y$. Hence, different skills emerge solely from the video-based reward $\mathcal{R}(s_t, a_t)$.

The policy receives proprioceptive measurements $s^{\text{proprio}}$ from the robot, including the positions and velocities of all 12 joints, as well as the orientation and angular velocity of the robot's base. Additionally, the previous action $a_{t-1}$ is provided as input. The skill command $y$ is input to the policy using a one-hot encoding.

Table 4: List of constraints employed during RL.

| Constraint | | Expression |
|---|---|---|
| Knee collision | $\mathcal{C}^{\text{knee collision}}$ | $= 1_{\text{knee collision}}$ |
| Base collision | $\mathcal{C}^{\text{base collision}}$ | $= 1_{\text{base collision}}$ |
| Foot contact force | $\mathcal{C}^{\text{foot contact}_j}$ | $= \|f^{\text{foot}_j}\|_2 - f^{\text{lim}}$ |
| Foot air time | $\mathcal{C}^{\text{air time}_j}$ | $= t^{\text{des}}_{\text{air time}} - t_{\text{air time}_j}$ |
| Joint limits (min) | $\mathcal{C}^{\text{joint}_k^{\min}}$ | $= q_k^{\min} - q_k$ |
| Joint limits (max) | $\mathcal{C}^{\text{joint}_k^{\max}}$ | $= q_k - q_k^{\max}$ |
| Joint velocity | $\mathcal{C}^{\text{joint velocity}_k}$ | $= |\dot{q}_k| - \dot{q}^{\text{lim}}$ |
| Joint acceleration | $\mathcal{C}^{\text{joint acceleration}_k}$ | $= |\ddot{q}_k| - \ddot{q}^{\text{lim}}$ |
| Torque | $\mathcal{C}^{\text{torque}_k}$ | $= |\tau_k| - \tau^{\text{lim}}$ |
| Action rate | $\mathcal{C}^{\text{action rate}_k}$ | $= \frac{|a_{t,k} - a_{t-1,k}|}{dt} - \dot{a}^{\text{lim}}$ |
| Base orientation around roll-axis | $\mathcal{C}^{\text{ori}_{\text{roll}}}$ | $= |\text{ori}_{\text{roll}}| - \text{ori}_{\text{roll}}^{\text{lim}}$ |

## C ADDITIONAL ILLUSTRATION OF THE ANIMAL KINGDOM DATASET

Figure 7 presents additional illustrations of videos selected from the Animal Kingdom dataset (Ng et al., 2022). In total, we used $8,791$ videos as our training dataset for our video-based reward function

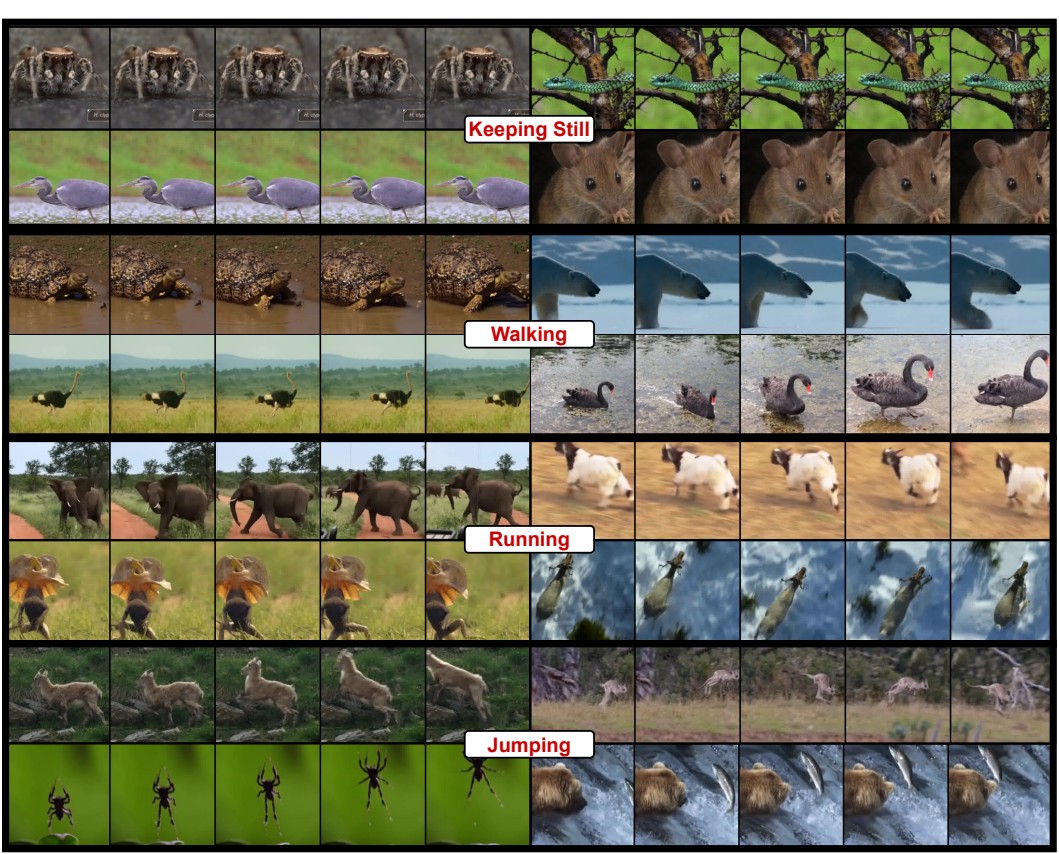

Figure 7: Additional illustrations of videos from the Animal Kingdom dataset (Ng et al., 2022) used to train the video-based reward function.

