# OpenReview forum: "Reinforcement Learning from Wild Animal Videos"
_ICLR.cc/2025/Conference — Submitted to ICLR 2025_

### Official Review · Reviewer_voL6 · 2024-11-04

**Soundness:** 2
**Presentation:** 3
**Contribution:** 3
**Rating:** 6
**Confidence:** 4

**Summary:**

This paper introduces an interesting idea to learn reward function for locomotion through wild animal video. The authors curate a dataset of 8.7 wild animal videos, train a video classifier and then use it as a reward model to train RL policy to control robot for locomotion. The multi-skill policy can be trained in a physical simulator and transfer to real world.

**Strengths:**

1. The idea is novel.
2. The paper is well written. Easy to follow.
3. Experiments and ablation among its own algorithm shows effectiveness of the proposed method.

**Weaknesses:**

1. It seems the paper lacks comparison to some baseline or other works. For example, can we compare the results in sim w/ some hand crafted reward models? Then you can compare sample efficiency of the proposed method.
2. Would like to know how large the animal dataset needs to be to make it work. This work uses 8.7K videos. Do we need more or it can work w/ less? Can we add an ablation on it?

**Questions:**

The paper claims that the reward model can help policy learn well in simulation and then successfully transfer to real. However, to me the "transfer to real" part seems orthogonal to the reward model itself. Could author explain why better reward model can lead to better sim-to-real transfer? For example, if we use a hand crafted reward function in the same setup and learn a policy in sim, can it also transfer to real? My impression is the answer should be yes.

---

> ### Author Response · Authors · 2024-11-21
>
> Thank you for your helpful feedback.
>
> We have prepared a short [Rebuttal_Video.mp4](https://www.dropbox.com/scl/fi/60glxqu1yjtlzusm8zv6r/Rebuttal_Video.mp4?rlkey=y1dx11pbbwvenst18a2sgm32k&st=72svyiw2&dl=0).
> This video aims to better illustrate our results, highlight the failure cases of our approach, and clarify the challenges of our problem setup.
> We kindly encourage you to watch it, as it may provide a fresh perspective on our work.
>
> ---
>
> We believe that the main contribution and significance of our paper may not have been fully recognized. Our work presents **the first successful demonstration of cross-embodiment visual imitation from a large, diverse and noisy internet dataset of animal video to physical robots**. The cross-embodiment capability arises from the diversity of animals in the dataset, making the video reward agnostic to the shape of the robot. This feature is uniquely enabled by using videos of animals in the wild, despite the underlying challenges discussed in the paper. These videos do not offer a proper physical grounding, which is then provided through reinforcement learning, leveraging a physics simulator and constraints that represent the physical limits of the robot. **Evidence of cross-embodiment is demonstrated in the experimental results**, such as the transfer of limb movements in the walking task. Our proof of concept underscores the vast potential of leveraging internet videos to advance locomotion capabilities.
>
> We address the rest of your concerns below.
>
> ---
>
> **Q. It seems the paper lacks comparison to some baseline or other works. For example, can we compare the results in sim w/ some hand crafted reward models? Then you can compare sample efficiency of the proposed method.**
>
> **A.** Regarding sample efficiency, we match the number of training epochs used in [1] while utilizing half the number of parallel environments in their setup for learning walking on flat ground with an Anymal robot. Manually designing rewards for each skill, such as in [1], would indeed outperform our approach which does not have direct access to ground truth rewards. Please note that tuning such ground truth reward functions for new skills is not trivial and can be time-consuming. However, the goal and contribution of our work are fundamentally different. We demonstrate that it is possible to transfer locomotion skills from a large, diverse, and noisy dataset of wild animal videos to physical robots. While our results are currently limited to simple skills, **we argue that our proof of concept highlights the potential of systematically leveraging video data for locomotion tasks, offering better scalability** compared to manual reward design.
>
> ---
>
> **Q. Would like to know how large the animal dataset needs to be to make it work. This work uses 8.7K videos. Do we need more or it can work w/ less? Can we add an ablation on it?**
>
> **A.** Thank you for the suggestion. In Appendix A, Figure 6, we provide an ablation study on the impact of dataset size on the learned behaviors. The results show that our approach works with slightly less data (e.g., 50% of the full dataset), but reducing the dataset size too much (e.g., 25% or less) prevents some target skills from emerging.
> This underscores the importance of training the video-based reward on a sufficiently large and diverse dataset to ensure effective generalization to robot locomotion.
>
> ---
>
> **Q. The paper claims that the reward model can help policy learn well in simulation and then successfully transfer to real. However, to me the "transfer to real" part seems orthogonal to the reward model itself. Could author explain why better reward model can lead to better sim-to-real transfer? For example, if we use a hand crafted reward function in the same setup and learn a policy in sim, can it also transfer to real? My impression is the answer should be yes.**
>
> **A.** We don’t make this claim in the paper. On the contrary, we argue that **transferring locomotion skills from animal videos to a real robot requires physical grounding**. This is addressed in Stage 2 of our approach, where we train a policy in a physics simulator using constrained RL. During this phase, the policy maximizes our video-based reward while adhering to behaviors that are physically plausible and transferable to a real robot.  We actually believe that most rewards, including our complex video-based reward, could be physically grounded through our setup. We have updated the paper to better emphasize this physical grounding aspect.
>
> ---
> 1. Learning to Walk in Minutes Using Massively Parallel Deep Reinforcement Learning, 2021

---

> > ### Author Response · Authors · 2024-11-25
> >
> > We hope that our new experiments and explanations addressed your concerns and are eager to hear your opinion before the end of the rebuttal.

---

> > > ### Comment · Reviewer_voL6 · 2024-11-26
> > >
> > > Thanks, the reply addresses my concerns. I would like to raise the score to 6.

---

### Official Review · Reviewer_zazf · 2024-11-04

**Soundness:** 2
**Presentation:** 4
**Contribution:** 2
**Rating:** 5
**Confidence:** 4

**Summary:**

This paper introduces Reinforcement Learning from Wild Animal Videos (RLWAV), a novel method for training quadruped robots to perform locomotion skills by observing wild animal videos. The authors train a video classifier on a large dataset of labeled animal videos and use the classification scores to provide rewards for RL training. RLWAV avoids the need for reference trajectories or hand-designed rewards by transferring learned skills from animals to robots. The method is validated both in simulation and on a real quadruped robot (Solo-12), demonstrating the transfer of skills such as walking and jumping.

**Strengths:**

1. Learning quadruped robot locomotion skills from existing wild animal locomotion is a good inspiration.
2. The task setup and experimental details are described clearly in the paper.

**Weaknesses:**

1. The current ablation study of the classifier training set is inadequate, making it hard to determine whether the method effectively utilizes cross-embodiment skills acquired from a diverse range of wild animal videos. The ablation should encompass factors such as the size of the training set and the number of different types of animals included in it.
2. While we anticipate gaining insights into four-legged movement skills from wild animal datasets, the only information we can provide the robot is the output of a classifier. This classifier appears to be able to achieve its task by focusing primarily on the background and the animal's torso, neglecting the movement of the four legs.

**Questions:**

1. If we do not use the trained classifier as a rewarder, but instead manually assign simple rewards to encourage the quadruped robot to stay still, walk forward, or jump, how effectively can the robot learn these skills?
2. How would the method perform if the output criterion of the classifier is only the movement of the robot's center of mass?

---

> ### Author Response · Authors · 2024-11-21
>
> Thank you for your helpful feedback.
>
> We have prepared a short [Rebuttal_Video.mp4](https://www.dropbox.com/scl/fi/60glxqu1yjtlzusm8zv6r/Rebuttal_Video.mp4?rlkey=y1dx11pbbwvenst18a2sgm32k&st=72svyiw2&dl=0).
> This video aims to better illustrate our results, highlight the failure cases of our approach, and clarify the challenges of our problem setup.
> We kindly encourage you to watch it, as it may provide a fresh perspective on our work.
>
> ---
>
> We believe that the main contribution and significance of our paper may not have been fully recognized. Our work presents **the first successful demonstration of cross-embodiment visual imitation from a large, diverse and noisy internet dataset of animal video to physical robots**. The cross-embodiment capability arises from the diversity of animals in the dataset, making the video reward agnostic to the shape of the robot. This feature is uniquely enabled by using videos of animals in the wild, despite the underlying challenges discussed in the paper. These videos do not offer a proper physical grounding, which is then provided through reinforcement learning, leveraging a physics simulator and constraints that represent the physical limits of the robot. **Evidence of cross-embodiment is demonstrated in the experimental results**, such as the transfer of limb movements in the walking task. Our proof of concept underscores the vast potential of leveraging internet videos to advance locomotion capabilities.
>
> We address the rest of your concerns below.
>
> ---
>
> **Q. Ablation study of the classifier training and cross-embodiment**
>
> **A.** The variations in the learned locomotion skills stem entirely from our video-based reward, which is trained exclusively on animal videos spanning hundreds of species, without any robot videos. There are no skill-specific rewards or reference trajectories, nor a transfer through a specific intermediate space (like configuration space, as used in other papers), making the emergent skills a direct result of cross-embodiment visual imitation. In Appendix A Figure 6, we present an ablation study on the impact of dataset size on the learned behaviors. The results show that significantly reducing the dataset size (e.g., to 25% or less) disrupts the proper emergence of locomotion skills. This underscores the importance of training on a sufficiently large and diverse dataset to ensure effective generalization to robot locomotion.
>
> ---
>
> **Q. Classifier focusing primarily on the background and the animal's torso, neglecting the movement of the four legs / How would the method perform if the output criterion of the classifier is only the movement of the robot's center of mass?**
>
> **A.** We disagree that the classifier neglects leg movements: while jumping may primarily rely on torso movement, this does not apply to other skills. Indeed, the camera tracks the torso in xyz and around the yaw axis, meaning that distinguishing between keeping still and walking/running depends solely on leg movements. Moreover, *Rebuttal_Video.mp4* highlights a failure case where, when commanded to walk, the robot moves only its front right and hinge left legs with walking-like motions while keeping the other two legs idle, effectively deceiving the video classifier. This shows that recognizing leg movements is crucial for achieving walking and running. On the contrary, explicitly extracting specific criteria (such as the center of mass), would make it more difficult to use videos of animals from the wild and could hinder the transfer process. This is because the diverse and unstructured nature of the animal videos requires a more general approach to enable successful cross-embodiment transfer.
>
> ---
>
> **Q. If we do not use the trained classifier as a rewarder, but instead manually assign simple rewards to encourage the quadruped robot to stay still, walk forward, or jump, how effectively can the robot learn these skills?**
>
> **A.** The tasks we considered in the paper are classical in legged locomotion, yet we are displaying a similar sample efficiency as other methods, such as [1] in their setup for learning walking on flat ground with an Anymal robot. Manually designing rewards for each skill, such as in [1],  would indeed outperform our approach which does not have direct access to ground truth rewards. Please note that tuning such ground truth reward functions for new skills is not trivial and can be time-consuming. However, the goal and contribution of our work are fundamentally different. We demonstrate that it is possible to transfer locomotion skills from a large, diverse, and noisy dataset of wild animal videos to physical robots. While our results are currently limited to simple skills, **we argue that our proof of concept highlights the potential of systematically leveraging video data for locomotion tasks, offering better scalability** compared to manual reward design.
>
> ---
>
> 1. Learning to Walk in Minutes Using Massively Parallel Deep Reinforcement Learning, 2021

---

> > ### Author Response · Authors · 2024-11-25
> >
> > We hope that our new experiments and explanations addressed your concerns and are eager to hear your opinion before the end of the rebuttal.

---

> > ### Comment · Reviewer_zazf · 2024-11-26
> >
> > Thank the authors for their response. The rebuttal makes the paper more clear. However, I still found it insufficiently evidenced that the method can truly learn cross-embodiment skills. Therefore, I have decided to maintain my score.

---

> ### Author Response · Authors · 2024-11-27
>
> We appreciate your careful consideration of cross-embodiment learning but firmly disagree with your assessment.
>
> Cross-embodiment imitation involves "agents learn policies from videos of other agents demonstrating the same task, but with stark differences in their embodiments" [1]. Our results demonstrate an unprecedented case of this, as shown in [this video capsule](https://www.dropbox.com/scl/fi/3pa6xko9qfdt7nsxl1x29/Rebuttal_2.mp4?rlkey=04k1kox2vq0hgoehkzj30nvnw&st=wocetxku&dl=0), with emergent skills arising solely from the transfer of animal video behaviors by design.
>
> Nonetheless, we thank the reviewer for the time and effort dedicated to reviewing our paper.
>
> ---
>
> 1. XIRL: Cross-embodiment Inverse Reinforcement Learning, 2021

---

### Official Review · Reviewer_BwRh · 2024-11-06

**Soundness:** 2
**Presentation:** 3
**Contribution:** 2
**Rating:** 5
**Confidence:** 4

**Summary:**

The paper trains a supervised video classification model on a dataset of wild animal videos (walking, running, standing, and jumping). It then uses the video model classifications as rewards to train a policy to control a quadroped robot in simulation. The policy is then transferred onto a quadroped robot in the real world.

**Strengths:**

- The paper studies an interesting problem of learning reward models from videos
- The proposed approach is interesting and in a good direction
- The paper is well written and the presentation is clear

**Weaknesses:**

- Position of the paper (title, abstract, intro) is a bit misleading. It suggests that the reward function would come purely form videos. However, the approach uses a number of hand-designed reward terms such as air time, symmetry, and terminations. I think that this is ok but the positioning of the paper should be updated to reflect that. In the current version of the approach, the video model serves only as part of the overall reward function.
- The results are promising but overall limited. Looking at the supplementary materials video it looks like the learnt skills do not quite match the desired behaviors, "keeping still" seems to be moving and "running" does not seem to be running.
- It would be good to ablate the impact of each of the reward terms. The current version of the manuscript includes the symmetry loss ablation which shows that the symmetry term plays a considerable role.

**Questions:**

Please see above.

---

> ### Author Response · Authors · 2024-11-21
>
> Thank you for your helpful feedback.
>
> We have prepared a short [Rebuttal_Video.mp4](https://www.dropbox.com/scl/fi/60glxqu1yjtlzusm8zv6r/Rebuttal_Video.mp4?rlkey=y1dx11pbbwvenst18a2sgm32k&st=72svyiw2&dl=0).
> This video aims to better illustrate our results, highlight the failure cases of our approach, and clarify the challenges of our problem setup.
> We kindly encourage you to watch it, as it may provide a fresh perspective on our work.
>
> ---
>
> We believe that the main contribution and significance of our paper may not have been fully recognized. Our work presents **the first successful demonstration of cross-embodiment visual imitation from a large, diverse and noisy internet dataset of animal video to physical robots**. The cross-embodiment capability arises from the diversity of animals in the dataset, making the video reward agnostic to the shape of the robot. This feature is uniquely enabled by using videos of animals in the wild, despite the underlying challenges discussed in the paper. These videos do not offer a proper physical grounding, which is then provided through reinforcement learning, leveraging a physics simulator and constraints that represent the physical limits of the robot. **Evidence of cross-embodiment is demonstrated in the experimental results**, such as the transfer of limb movements in the walking task. Our proof of concept underscores the vast potential of leveraging internet videos to advance locomotion capabilities.
>
> We address the rest of your concerns below.
>
> ---
>
> **Q. Position of the paper is a bit misleading. It suggests that the reward function would come purely form videos. However, the approach uses a number of hand-designed reward terms such as air time, symmetry, and terminations.**
>
> **A.** In our paper, we argue that transferring locomotion skills from animal videos to a real robot requires physical grounding. This is addressed in Stage 2 of our approach, where a policy is trained in a physics simulator using constrained RL. During this phase, the policy maximizes the video-based reward while adhering to constraints that ensure physically plausible and transferable behaviors. These constraints are necessary for physical grounding, as many parameters—such as the torque limits of the Solo-12 robot—cannot be inferred from animal videos. During RL, the policy learns multiple skills simultaneously, but the constraints are applied uniformly across skills. Consequently, **the variations in the learned skills come purely from videos**. We have revised the paper to better highlight this critical aspect of physical grounding. Employing such constraints or similar penalties is a standard and essential practice in learning-based locomotion for real-world robots, regardless of the target locomotion skill, see [1, 2] for example. We anticipate that more advanced video models and more accurate physics simulators could eliminate the necessity for symmetry and air-time constraints.
>
> ---
>
> **Q. The results are promising but overall limited. Looking at the supplementary materials video it looks like the learnt skills do not quite match the desired behaviors.**
>
> **A.** We acknowledge the limitations of our results. However, our proof of concept demonstrates the potential of leveraging large video datasets sourced from the internet for locomotion tasks. This problem is particularly challenging due to the inherent diversity and noisiness of internet videos.
> - *Keeping Still*: While the robot's feet move slightly, it remains stationary without shifting its base. This aligns with many "keeping still" videos in the dataset, which often show animals exhibiting minimal in-place motion.
> - *Running*: We agree the resulting motion resembles trotting rather than running. However, the robot demonstrates broader and faster movements compared to walking, reflecting an intermediate gait.
>
> We refer to *Rebuttal_Video.mp4* for additional qualitative insights.
>
> ---
>
> **Q. It would be good to ablate the impact of each of the reward terms.**
>
> **A.** We have added an ablation for the base orientation constraint around the roll axis and the foot air-time constraint in Appendix A Table 3. In our *Rebuttal_Video.mp4*, we show that without the air-time constraint, the robot performs motions visually close to walking but instead slips in place by generating high-frequency ground contacts, exploiting imperfections in the simulator. Regarding symmetry loss, our *Rebuttal_Video.mp4* illustrates failure cases when it is removed. Without symmetry, the robot may deviate from straight running, turning instead while maintaining trotting motions. Additionally, instead of walking, it may keep two feet on the ground while simulating running motions with only two legs, failing to move the body forward but still deceiving the video classifier
>
> ---
>
> 1. Not Only Rewards But Also Constraints: Applications on Legged Robot Locomotion, 2024
> 2. Extreme Parkour with Legged Robots, 2024

---

> ### Author Response · Authors · 2024-11-25
>
> We hope that our new experiments and explanations addressed your concerns and are eager to hear your opinion before the end of the rebuttal.

---

> ### Comment · Reviewer_BwRh · 2024-11-27
>
> Thank you for the response, rebuttal video, and the additional ablations. I appreciate the acknowledgement of the results (Q2) and the videos of policies trained with and without the symmetry and air time rewards (Q3). I increased my score.
>
> My main remaining concern is related to Q1. I think that this is a promising and interesting direction. However, in its current form, I feel that there is a mismatch between the paper claims and results. My suggestion is to keep working on improving the method to get to a level of performance that better substantiates the claims (namely, not reliant on hand-coded auxiliary rewards like air time and symmetry & results that better reflect the desired skills like keeping still or running) or to soften the claims to better reflect the current level of performance (video classifiers serving as an auxiliary reward & preliminary transfer results).

---

> ### Author Response · Authors · 2024-11-28
>
> Thank you for your thoughtful response.
>
> We acknowledge that symmetry and air time are expedients. However, these do not detract from the generality of our results, particularly within the scope of existing research in this area. Below, we provide essential context that could shift your perspective on the significance of our results, justifying better the positioning of our paper. We also tried to better adjust our wording to match your recommendation.
>
> ---
>
> **1. Imitating wild animal videos is harder than you seem to realize.**
>
> For clarity, we included only some of the most identifiable animal videos in the rebuttal video, which may underplay the difficulty of our setting. However, the videos are generally noisier and the actions unclear. To illustrate this, we compiled some other videos used to train our reward model [in this video](https://www.dropbox.com/scl/fi/h31t05ekwhyiul9b008nf/AK_compilation.mp4?rlkey=uhj1mgpittjqv4fdh8ixzb2w1&st=oq0e3sey&dl=0). Viewing these should make the challenges evident and underscore the impressiveness of our results.
>
> Compared to other works on visual imitation from large video datasets, the most relevant (large cross-embodiment and deployment on a real robot) is perhaps [1], although it focuses on manipulation. Even though only a subjective comparison is possible, we believe our cross-embodiment setting is more pronounced, our dataset noisier, and our demonstrated skills more complex.
>
> While we do not claim to solve visual imitation—a challenge that will require many more breakthroughs—we firmly believe our paper represents a significant step forward, showcasing an unprecedented demonstration of extreme cross-embodiment transfer.
>
> ---
>
> **2. Skill-agnostic constraints are an integral aspect of RL locomotion, as they are unavoidable.**
>
> The constraints we used are ubiquitous in RL locomotion. [2, 3] for instance learn policies that can walk, crawl, leap and climb, all using a very similar set of core/skill-agnostic constraints as ours, yet they use additional skill-specific rewards and/or hand-designed terrains in simulation to ensure the emergence of the target skills.
>
> Indeed, these skill-agnostic constraints are obligatory: except for the three discussed below, the constraints we employ address robot-specific limitations, such as torque, or velocity limits, making them mandatory yet not inferable from animal data.
>
> - *Air time*: While not ideal, this constraint is widely used (ex: [2–6]). One reason for this is the limitations in physics simulators, which assume rigid bodies. This leads to unrealistic behaviors, as explained in our previous reply. Addressing this via video-based rewards would require much finer motion understanding, which is a significant challenge (see above).
> - *Symmetry regularization*: Also used in [7] for example, this constraint aligns with the natural symmetry of quadrupeds and their locomotion patterns. Our reward model observes the robot from one side, necessitating symmetry to ensure robustness (as shown in the Rebuttal Video).
> - *Base orientation*: has negligible impact (see ablation in the paper)
>
> ---
>
> **3. Modification in the paper to soften the claims**
>
> We will revise the last part of the abstract to clarify that the skills emerge “*without relying on reference trajectories nor* ***skill-specific*** *rewards.*”, consistent with our modifications to the introduction. We carefully revised the introduction and found no over-claim following your remark. We will change the “constraints” paragraph of section 3.3 to be more explicit about the importance of auxiliary constraints. We will modify the conclusion to explicit this issue in the limitation.
>
> We hope that, with those arguments in mind, you will now agree with our wordings of the claims.
>
> ---
>
> 1. Learning Generalizable Robotic Reward Functions from "In-The-Wild" Human Videos, 2021
> 2. Extreme Parkour with Legged Robots, 2024
> 3. SoloParkour: Constrained Reinforcement Learning for Visual Locomotion from Privileged Experience, 2024
> 4. Learning to Walk in Minutes Using Massively Parallel Deep Reinforcement Learning, 2021
> 5. Learning Agile Skills via Adversarial Imitation of Rough Partial Demonstrations, 2022
> 6. Legs as Manipulator: Pushing Quadrupedal Agility Beyond Locomotion, 2023
> 7. Not Only Rewards But Also Constraints: Applications on Legged Robot Locomotion, 2024

---

### Meta-Review · Area_Chair_PqzQ · 2024-12-21

**Metareview:**

The paper introduces Reinforcement Learning from Wild Animal Videos (RLWAV), which uses a video classifier trained on large-scale wild animal footage to reward a quadruped robot’s locomotion policy. By combining classifier scores with a few hand-designed reward terms, the policy learns walking, jumping, and standing behaviors in simulation and then transfers them to a real Solo quadruped. The key claim is that reference trajectories are unnecessary because animal videos provide diverse, natural examples of movement.

Strengths:

-- Clear Presentation: Methodology and experiments are explained coherently.

-- Real-World Transfer: Demonstrates feasibility on an actual quadruped robot.

Weakness:

-- Incomplete Positioning: Still relies on additional hand-crafted rewards (e.g., symmetry), which is underemphasized.

-- Limited Ablations: Lacks thorough studies on dataset size, animal variety, and reward component importance.

-- Lack of Comparisons: No strong baselines or simpler reward models for fair benchmarking.

After carefully reading the paper, the reviews and rebuttal discussions, the AC find while the concept is innovative, the results are somewhat limited. The AC agrees with the majority of reviewers on rejecting the paper.

**Additional Comments On Reviewer Discussion:**

See the weakness and comments above, while some of the reviews' concerns are addressed, there are still remaining concerns.

---

### Decision · Program_Chairs · 2025-01-22

Reject